# Mortality Caused by *Candida auris* Bloodstream Infections in Comparison with Other Candida Species, a Multicentre Retrospective Cohort

**DOI:** 10.3390/jof9070715

**Published:** 2023-06-29

**Authors:** Cynthia Ortiz-Roa, Martha Carolina Valderrama-Rios, Sebastián Felipe Sierra-Umaña, José Yesid Rodríguez, Gerardo Antonio Muñetón-López, Carlos Augusto Solórzano-Ramos, Patricia Escandón, Carlos Arturo Alvarez-Moreno, Jorge Alberto Cortés

**Affiliations:** 1Department of Internal Medicine, Faculty of Medicine, Universidad Nacional de Colombia, Bogotá 111321, Colombia; 2Subred Integrada de Servicios de Salud Sur E.S.E. Hospital Tunal, Bogotá 110621, Colombia; 3Clínica Integral de Emergencias Laura Daniela, Instituto Cardiovascular del Cesar, Centro de Investigaciones Microbiológicas del Cesar (CIMCE), Valledupar 200001, Colombia; 4Subred Integrada de Servicios de Salud Suroccidente E.S.E. Hospital Suroccidente Kennedy, Bogotá 110871, Colombia; 5Grupo de Microbiología, Instituto Nacional de Salud, Bogotá 111321, Colombia; 6Infectious Diseases Unit, Hospital Universitario Nacional, Bgootá 111321, Colombia

**Keywords:** *Candida auris*, candidemia, bloodstream infection, antifungals, intensive care, Colombia

## Abstract

*Candida auris* is an emerging pathogen considered to be critical in the World Health Organization fungal organisms list. The study aims to determine the mortality and hospital stays attributed to *Candida auris* (*C. auris*) compared to other *Candida* species in adult patients with candidemia. A retrospective cohort of adults with candidemia was examined from seven centres in Colombia between 2016 and 2021. The primary outcome was 30-day mortality, and the secondary outcome was the length of hospital stay among survivors. Adjustment of the confounding variables was performed using inverse probability weights of exposure propensity score (candidemia by *C. auris*), survival regression models (Weibull distribution), and a counting model (negative binomial distribution). A value of 244 (47.6%) of the 512 patients with candidemia died within the first 30 days. The crude mortality in *C. auris* was 38.1% vs. 51.1% in *Candida* non-auris (CNA). In the Weibull model, mortality in the *C. auris* group was lower (adjusted HR: aHR- 0.69, 95% CI: 0.53–0.90). Antifungal treatment also decreased mortality, with an aHR of 0.36 (95% CI 0.27–0.47), while the presence of septic shock on patient progression increased it, with an aHR of 1.73 (95% CI 1.41–2.13). Among the patients who survived, no differences in the length of hospital stay were observed between the *C. auris* and the CNA groups, with an incidence rate ratio of 0.92 (95% CI: 0.68–1.22). Mortality in patients with *C. auris* bloodstream infections appears lower when adjusted for numerous confounding variables regarding treatment and the presence of septic shock in patient progression. We identified no significant effect of *C. auris* on the length of hospital stay in surviving patients.

## 1. Introduction

Candidemia is one of the most frequent infections in intensive care units, with an associated mortality of more than 30%, which may be higher in patients with septic shock [1]. *C. auris* is a species first described in Japan in 2009 [2], and from 2011, sporadic cases and outbreaks of mainly bloodstream infections from this pathogen began to appear in different geographical regions. *C. auris* differs from the other identified species of *Candida* by its greater ease of cross-transmission [3] and resistance to antimycotics [4]. The first cases of infection reported simultaneously in the world occurred in five countries: Japan, Pakistan, India, South Africa, and Venezuela [5]. The World Health Organization (WHO) recently placed this species among the priority fungal pathogens to guide research, development, and action in public health [6]. In Colombia, *C. auris* infections are a mandatory notification event according to the National Alert issued by Instituto Nacional de Salud de Colombia (in English, the National Institute of Health) in 2016 [7].

*C. auris* infections represent a challenge for clinicians due to difficulties in their microbiological identification, their profile of resistance to antifungals, and their ability to persist and spread through surfaces [8]. The crude mortality rate in *C. auris* infections appears to be comparatively higher than for other species of *Candida,* ranging from 33% to 72% [9]. A systematic review and meta-analysis that included cases between 2009 and 2019 from different countries reported an average crude mortality of 45% (95% CI: 39–51%) for *C. auris* bloodstream infections [10]. However, mortality attributable to *C. auris* remains unclear. It has not been demonstrated that patients with invasive *C. auris* infections are less likely to die than patients infected with other species of *Candida* [11]. The comparative data are limited, considering that frequently patients who develop candidemia present multiple comorbidities that affect the prognosis, representing potential confounding factors. Therefore, in addition to the problems associated with multidrug resistance and the ability to generate outbreaks, the clinical impact on individual patients is unclear.

The present study aims to determine the mortality of patients with *C. auris* bloodstream infections compared to other *Candida* species through a cohort of adult patients, adjusted by analyzing inverse weights of the propensity score.

## 2. Materials and Methods

### 2.1. Study Design and Population

A multicentre retrospective cohort study was conducted. We included cases of patients with candidemia aged 18 years or older with at least one positive blood culture for any species of *Candida* in seven tertiary referral hospitals in Colombia, located in Bogotá D.C. and Valledupar. We invited these centres to participate in the study because they are referral centres, which due to their complexity, have identified a relatively high frequency of *Candida* species infections, with adequate monitoring of *C. auris* in their laboratory, and which usually treat high-risk patients for fungal infections such as oncology patients, abdominal surgery patients, and extended stay in Intensive Care Unit (ICU), among others. For the selection of potential patients to be included, the information reported by each centre’s laboratories was used and filtered through the WHONET 5.6 software (WHO) to identify episodes of *Candida* bloodstream infections from 2016 to 2021. The cases of identified candidemia by laboratory report were reviewed. The cases corresponding to the first positive blood culture for *Candida* species were included, obtaining the definitive list of patients to be included in the data collection. We excluded the data of repeated patients, patients with no available electronic medical history, isolates corresponding to samples other than blood cultures, and different yeasts besides *Candida* species in the final identification. We included all patients who did not receive antifungal treatment.

The medical records of each identified case were reviewed, and the information of interest was recorded in an online form using REDCap (Vanderbilt University, USA). The sociodemographic, clinical, and paraclinical variables were collected from the review of the medical records available in electronic media and were recorded in the data collection format.

The participating centres in Bogotá were Hospital Universitario Nacional de Colombia (HUN); Hospital Simón Bolívar of the Subred Integrada de Servicios de Salud Norte E.S.E. (HSB); Hospital de Suroccidente Kennedy of the Subred Integrada de Servicios de Salud Suroccidente E.S.E. (HOK); and Hospital El Tunal of the Subred Integrada de Servicios de Salud Sur E.S.E (HET). Valledupar’s centres were Clínica Integral de Emergencias Laura Daniela (CELD), Instituto Cardiovascular del Cesar (ICVC), and Clínica Alta Complejidad del Caribe (CACC).

Appropriate treatment was considered if it started between days 0 and day +2. The date of taking the blood culture was counted as day 0, and late treatment was administered from day +3. Immediate mortality corresponds to deaths occurring between day 0 and day +2, early mortality between day 0 and day +7, and late mortality between day +8 to day +30. Stay in the ICU was considered if candidemia had occurred within 14 days before or after admission to the ICU.

### 2.2. Microbiological Information

The data derived from the processing of blood culture samples, the detection of candidemia, and the identification of the *Candida* species were taken from the laboratory records of the participating centres, whose processes were carried out in line with the protocols of each centre, according to the available methods of microbiological identification for the period included in the study, between 2016 and 2021. Antifungal susceptibility test data were not included due to their unavailability in all centres during the study period.

A proportion of *C. auris* isolates was confirmed in the Microbiology Laboratory of Instituto Nacional de Salud de Colombia (INS) by MALDI-TOF MS Biotyper (Bruker Daltonics, Billerica, MA, USA) MBT version 4.1.80 or PCR methods, or both, considered as the gold standard for the identification of *C. auris* [12,13]. The isolates identified in Valledupar were confirmed using MALDI-TOF MS technology, and complementarily the samples were sent to the University of Nantes in France for sequencing [14,15]. In Bogotá, one of the participating centres has had MALDI-TOF MS technology as a microbiological identification method since 2017 (HOK), and the other 3 centres used phenotypic identification methods such as MicroScan Walkaway^®^ (HET) and BD Phoenix^®^ (HET, HUN, HSB).

### 2.3. Exposure

The identification of *C. auris* was considered as exposure. Isolates of *C. auris* included those samples initially identified as *C. auris*, *C. auris/haemulonii*, and *C. haemulonii* [14]. Of the isolates included in the study as *C. auris*, 58 were sent to the INS (Colombian National Institute of Health), of which 56 (96.6%) were confirmed as *C. auris*, and 2 were discarded and identified as other *Candida* species. The latter two cases were assigned to the Candida non-auris (CNA) group according to their final identification. Three patients were identified in the first blood culture as CNA (*C. albicans*, *C. utilis*, *C. parapsilosis*); however, in the follow-up blood culture, *C. auris/haemulonii* were identified, and they were considered as mixed candidemia and assigned to the *C. auris* group. The control group corresponded to those identifying any *Candida* isolation other than *C. auris* (CNA).

### 2.4. Outcomes

The primary outcome was mortality at 30 days from a *Candida* species’ first positive blood culture. Secondary was the length of the hospital stay, considered in patients who had survived during the available follow-up time.

### 2.5. Statistical Analysis

Descriptive statistics were performed with relative and absolute frequencies, and the means and their standard deviation, and the medians and quartiles 25 to 75, were calculated according to their distribution. For some missing data, imputations were performed using the mean or expected values according to the clinical context.

An analysis was carried out using weights of the inverse probability of exposure obtained through propensity scores; it allowed us to balance the variables in the exposed and control groups similarly. A propensity-to-be-exposed score was calculated to control for confounding variables using a logistic regression model, for which the outcome was *C. auris* bloodstream infections. A value of 27 variables was included in the model: previous stay in the ICU; serum creatinine; the ratio between the fraction of inspired and oxygen blood pressure; the fraction of inspired oxygen; organic dysfunction; sepsis; dialytic support; parenteral nutrition; SOFA score of sepsis; use of vasoactive medications; mechanical ventilation before isolation; previous surgery (3 months); use of previous antifungal; previous antibiotic use; previous bacteremia; time to candidemia (from admission); Charlson comorbidity score; the presence of cancer, diabetes, chronic kidney disease, lung disease, cardiovascular disease, COVID-19; age; gender; and the two hospitals that contributed the most significant number of patients (HET, HSB). Exposed individuals were assigned a weight of 1/propensity score x prevalence of exposure, while unexposed individuals were assigned a weight equivalent to 1/(1-propensity score) x prevalence of non-exposure. The balance of the variables was assessed visually by plotting their distribution and calculating the standardized mean deviations (SMD) after weight allocation.

Subsequently, mortality was compared using a Weibull survival parametric regression model, taking into account that the variables did not meet the proportionality assumption. The outcome (dependent) variable was death at 30 days, and the predictor variable was candidemia for *C. auris*. Adjustments were made for potentially confounding variables of events that occurred after the identification of candidemia. Comparisons were made using Kaplan-Meier curves utilizing the pseudo population with inverse weights and the Wald test (χ^2^). A counting model with a negative binomial distribution was used for the hospital stay outcome. For the results, a *p* < 0.05 was taken into account to be considered statistically significant, and robust errors were used for the reporting of confidence intervals.

## 3. Results

In total, 512 cases of candidemia were identified, 134 (26.1%) of *C. auris* and 378 of (73.9%) CNA (Figure 1). Table 1 shows the frequency of the different *Candida* species identified. Among others are four isolates of *Candida* species, two of *C. melibiosica*, two of *C. guilliermondii*, and one of each of the following species, respectively: *C. firmetaria*, *C. lusitaniae*, *C. pelliculosa*, *C. rugosa*, and *C. sake*. Three cases of mixed candidemia with *C. auris* were identified, each with an isolation of *C. albicans*, *C. parapsilosis*, and *C. utilis*. The comparison of the characteristics of each group is shown in Table 2.

The time for reporting the final identification of the *Candida* species was similar for the *C. auris* vs. CNA; both had a median of 4 days (IQR: 3–6) (Table 1). The time to candidemia in each group is shown in Table 2. In the *C. auris* group, 83 cases (62%) had at least one control blood culture, while in the CNA group, 154 cases (41.2%) had it with a median blood culture intake control of 7 days (IQR: 7–10) in both the exposed and unexposed groups. Among patients with control blood culture, a statistically significant difference in microbiological persistence was found, being greater in the *C. auris* group, with 15 cases (18%) vs. 16 cases (10%) in the CNA group (*p* = 0.007).

A total of 166 (32.4%) patients with COVID-19 were included, of which 145 (28.3% of the total) patients received corticosteroids before candidemia, 40 (29.9%) in the *C. auris* group and 105 (27.8%, *p* = 0.72) in the other group.

### 3.1. Treatment and Mortality

A total of 441 patients (86.1%) received antifungal treatment during the candidemia episode, and 175 (34.1%) received it promptly (Table 3). Likewise, before the episode of candidemia, 55 (10.7%) patients received an antifungal during hospitalization. Caspofungin was administered in 204 patients, 88 (65.7%) cases with *C. auris* and 116 (30.7%) with CNA (*p* < 0.001); fluconazole was used in 245 patients, 34 (25.4%) in *C. auris* and 211 (55.8%) in CNA (*p* < 0.001). Differences between the timely use of caspofungin were also observed: 39 cases (29.1%) in *C. auris* vs. 51 cases (13.5%) in CNA (*p* < 0.001).

Of the total number of patients, 288 (56.2%) died, 244 (47.6%) did so in the first 30 days. Deaths occurred according to this distribution: 53 (18.4%) immediately, 69 (13.4%) early, and 122 (42.3%) late. The treatment distribution and unadjusted outcomes for each group are shown in Table 3.

After adjustment for propensity scores and other variables, the 30-day mortality for *C. auris* was lower than in the CNA group, with HR adjusted at 0.69 (95% CI 0.53–0.90). Lower mortality was found for those receiving treatment, with HR adjusted at 0.36 (95% CI 0.27–0.47), and mortality was higher in those with septic shock at evolution, with HR adjusted at 1.73 (95% CI 1.41–2.13). Figure 2 shows the result of adjustment by the Weibull multivariate survival model.

### 3.2. Hospital Stay

The average stay was 30 days (IQR: 18.75–49.25). The maximum discharge time was 253 days. In patients who survived, the median time to discharge was 32.5 days (IQR: 18.75–53.25) for the *C. auris* group and 28 days (IQR: 18.75–49.0) for the CNA group (*p* = 0.76). For the adjusted comparison of hospital stay, a negative binomial model was performed in which the 224 patients who survived, 68 (30.3%) from the *C. auris* group and 156 (69.6%) from the CNA group, were included. The effect of exposure (*C. auris*) on hospital stay was assessed using the adjusted incidence rate ratio and was 0.92 (95% CI: 0.69–1.22). The comparison was also adjusted for variables such as ICU admission with adjusted IRRs of 1.29 (95% CI: 0.98–1.70) and previous surgery with an adjusted IRR of 1.31 (95% CI: 0.95–1.81).

## 4. Discussion

In this retrospective cohort study, in seven hospitals in two cities of Colombia, it was found that mortality in the group of patients with *C. auris* was lower when compared with the CNA group after a balance by weights of the inverse probability of exposure obtained by propensity and adjustment scores for antifungal treatment and the presence of septic shock in the evolution. No statistically significant difference in hospital stay was identified between surviving patients from the two groups.

Previous studies have compared outcomes between patients with *C. auris* bloodstream infections and candidemia by other *Candida* species, finding similar unadjusted mortality between the two groups and significant differences in other unadjusted outcomes, such as a longer length of hospitalization and greater exposure to antifungals in the *C. auris* group [16,17]. A previous report that included some of the patients from this study showed a similar rate of mortality but failed to identify a statistical difference to the control group, most likely due to the limited number of cases and lack of statistical power [15]. Two retrospective studies in Pakistan found a similar crude death rate between the two groups, one including 38 cases of *C. auris* and 101 of CNA with a mortality of 62.6% vs. 52.5%, HR 1.45 (95% CI: 0.84–2.4) [16], and another study that included a smaller sample of candidemia in COVID-19 patients, 4 cases of *C. auris*, and 22 of CNA, finding a similar mortality rate 67% in *C. auris* vs. 65% CNA [17].

Lower mortality in the *C. auris* group after adjustment for multiple potentially confounding variables suggests that the pathogenicity of *C. auris* may be lower than that of other *Candida* species. A multicentre, case-control study in New York included 196 patients, 86 of whom had *C. auris* infections, and identified that a bloodstream infection was not associated with increased mortality at 30 or 90 days after an inverse propensity adjustment [18].

Although unique biological, genetic, epidemiological, resistance, and transmission characteristics are recognized in *C. auris,* multiple virulence factors are shared with *C. albicans* [19]. Comparative animal models of murine and invertebrates indicate that *C. auris* is less virulent than *C. albicans* [20]. However, a study conducted with Colombian isolates showed that *C. auris* strains recovered from colonisations demonstrates greater virulence in the invertebrate model *Galleria mellonella* compared to those isolates recovered from patients with bloodstream infections [21]. From these models, it has been proposed that the decrease in virulence could be related to the inability of *C. auris* to develop hyphae or pseudohyphae in mammals, which represents a fundamental role in the ability to invade tissues [22].

In the hospitals in our study, 26.1% of patients had *C. auris* isolation, which represents a change in the epidemiological profile observed in recent years and is consistent with a relative decrease in the frequency of *C. albicans* isolates, as well as the progressive increase in other species including *C. parapsilosis* and *C. glabrata* [23]. The epidemiological profile of our patients presents a lower frequency of some high-risk groups, such as major surgeries, total parenteral nutrition, cancer, advanced age, immunosuppressive treatments, and neutropenia [24], suggesting that the most critical factor for the prevention of a *C. auris* bloodstream infection could be related to better adherence to infection prevention and control measures in high complexity centres. Data obtained from outbreaks of *C. auris* infections have shown rapid colonisation from hospital admission and prolonged microbiological viability [25]; for this reason, recommended measures for the prevention of infections include handwashing, contact isolation, personal protection elements, obtaining information from the medical centre of origin for the case of referred patients, the isolation of patients with a history of *C. auris* infection who re-enter a health centre, and the centres’ cleaning and disinfection strategies, which have shown efficacy in controlling and preventing *C. auris* infections [26,27].

According to the results obtained, *C. auris* does not seem to have a significant effect on the length of hospital stay compared to other CNAs; this finding could be explained because patients with candidemia have a prolonged stay associated with multiple comorbidities and other concomitant pathogenic conditions that cannot be controlled with this type of methodological design. In our study, microbiological persistence was significantly higher in the *C. auris* group compared to CNA, 18% and 10%, respectively (*p* = 0.007), which coincides with that observed by Simon et al., who found an increased risk of microbiological recurrence within 60 days in the group of patients with *C. auris* bloodstream infections (aOR 4.46) [18].

Strengths of our study include a high sample size (relative to previously published studies), the inclusion of different high-complexity hospitals, and the use of a propensity analysis for inverse weights by adjusting for confounding variables. Among the weaknesses, we can mention that it was not possible to confirm 43.3% of the isolates of *C. auris (C. auris/haemulonii*), which were identified by the automated methods available in the centres, which do not provide the precision required for the correct identification of genus and species. Phenotypic methods ideally require a complementary identification method [13,28]; VITEK 2 has reported the best specificity of 88.2% [29]. The specificity of the identification of *C. auris* from the centres’ methods available in this study was 96.5% (in comparison to the identification made by the Colombian National Institute of Health).

The susceptibility profile was partially available in the included cases (data not included in the study). It is important to mention that implementing antifungal susceptibility testing in clinical practice could adequately help to guide treatment and impactfully address the current emergency of drug-resistant *Candida* species [30]. However, it still presents some limitations in its implementation, including correct interpretation, clinical cut-offs, time to results, and lack of reliable data in the case of some non-*albicans Candida*, such as *C. auris,* whose isolates must be tested and interpreted using clinical cut-offs derived from other *Candida* species [5].

Part of the cohort included patients with COVID-19. However, the effect of this variable was not visible, since this characteristic was included in the regression model used to construct the propensity score. In the pseudo population, no difference was noted in those with or without COVID-19. Neither did we collect data on bacterial or viral coinfections.

Our findings reinforce the importance of the early identification of infections by *Candida* species and laboratory surveillance concerning the opportunity for treatment initiation, establishing infection prevention and control measures, and their subsequent impact on morbidity and mortality. We observed a low use of echinocandins in the studied population despite what is recommended by evidence-based treatment guidelines [31,32] and the availability of this group of drugs in Colombia. Different studies have previously shown the impact of appropriate treatment of *Candida* species infections on mortality [33,34] and so it is considered relevant to reinforce the implementation of the centre’s interventions to ensure adherence to clinical practice guidelines.

## 5. Conclusions

Our study suggests that mortality in the *C. auris* bloodstream infection group is lower compared to CNA candidemia; however, crude mortality remains high (38.1%), consistent with data previously published in other previous studies, reporting data between 30% to 72% [9,10,18], indicating that efforts should be concentrated on the prevention of infections, early microbiological diagnosis, and the timely initiation of appropriate antifungal treatment, ideally guided by antifungal susceptibility.

## Figures and Tables

**Figure 1 jof-09-00715-f001:**
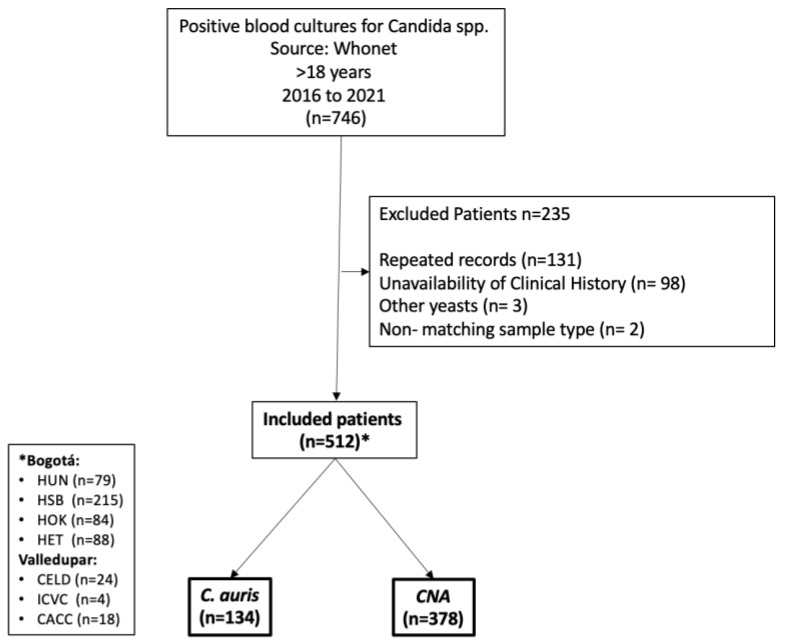
A flowchart of patient inclusion in the cohort. CNA: *Candida* non-*auris*; * Hospitals in Bogotá and Vadellupar: HUN: Hospital Universitario Nacional, HSB: Hospital Simón Bolívar, HOK: Hospital de Suroccidente Kennedy, HET: Hospital El Tunal, CELD: Clínica Integral de Emergencias Laura Daniela, ICVC: Instituto Cardiovascular del Cesar, CACC: Clínica Alta Complejidad del Caribe.

**Figure 2 jof-09-00715-f002:**
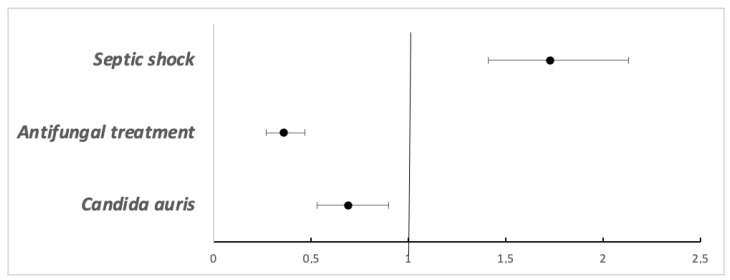
The hazard ratios of variables affecting mortality in the cohort of patients with candidemia. *n* = 512, Robust 95% CI.

**Table 1 jof-09-00715-t001:** The *Candida* species identified and reporting time across seven centres in Colombia, 2016–2021.

Species Identified	n (%)	Reporting Time Median in Days (IQR ^1^)
*C. auris*	134 (26.1)	4 (3–6)
CNA ^2^	378 (73.8)	
*C. albicans*	187 (36.5)	5 (3–6)
*C. parapsilosis*	97 (18.9)	4.5 (3–7)
*C. tropicalis*	45 (8.7)	4 (2–5)
*C. glabrata*	27 (5.3)	3 (2–5)
*C. dubliniensis*	5 (0.9)	
*C. krusei*	4 (0.7)	
Others	13 (2.5)	

^1^ IQR: Interquartile range. ^2^ CNA: Candida non-auris.

**Table 2 jof-09-00715-t002:** Demographic, clinical, and exposure characteristics of patients with candidemia in seven centres in Colombia, 2016–2021.

Characteristic	All n = 512	*C. auris* n = 134	CNA ^1^ n = 378	SMD ^2^
Male gender (%)	317 (61.9)	86 (64.2)	231 (61.1)	0.063
Age, years (mean (SD ^3^))	68.5 (17.7)	54.3 (17.1)	60.0 (17.6)	0.331
COVID-19 (%)	166 (32.4)	47 (35.1)	119 (31.5)	0.076
Congestive heart failure (%)	43 (8.3)	7 (5.2)	36 (9.5)	0.165
Cardiovascular Disease (%)	195 (38.0)	44 (32.8)	151 (39.9)	0.148
COPD (%)	66 (12.8)	12 (9.0)	54 (14.3)	0.167
Chronic kidney disease (%)	34 (6.6)	14 (10.4)	20 (5.3)	0.192
Diabetes (%)	97 (18.9)	30 (22.4)	67 (17.7)	0.117
Cancer (%)	32 (6.25)	13 (9.7)	19 (5.0)	0.180
Abdominal surgery (%)	58 (11.3)	17 (12.3)	41 (10.8)	0.319
Bacteremia 14 days prior (%)	127 (24.8)	40 (29.8)	87 (23.0)	0.155
Broad-spectrum antibiotic 14 days prior	477 (93.2)	132 (98.5)	345 (91.8)	0.333
Charlson comorbidity index (mean (SD))	2.6 (2.2)	2.4 (2.1)	2.7 (2.2)	0.184
SOFA (mean (SD))	7.96 (3.9)	7.42 (3.3)	8.15 (4.07)	0.196
Intensive Care Unit (%)	267 (52.2)	64 (47.8)	203 (53.7)	0.119
Time to candidaemia (mean (SD))	20.0 (16.0)	22.9 (16.4)	19.1 (15.9)	0.231
Referred from another hospital (%)	81 (15.8)	21 (15.7)	60 (15.9)	0.006
Mechanical ventilation (%)	316 (61.7)	83 (61.9)	233 (61.6)	0.006
Total parenteral nutrition (%)	120 (23.4)	33 (24.6)	87 (23.0)	0.038
Dialysis (%)	73 (14.2)	18 (13.4)	55 (14.6)	0.032
Hypotension (%)	48 (9.3)	15 (11.1)	33 (8.7)	0.082
Sepsis (%)	349 (68.1)	100 (74.6)	249 (65.8)	0.192
Septic shock (%)	140 (27.3)	32 (23.8)	108 (28.5)	0.107
Central venous catheter (%)	444 (86.7)	117 (87.3)	327 (86.5)	0.024
FiO2 (median (RIQ))	40 (32–50)	40 (32–50)	40 (32–50)	0.008
PaO2/FiO2 ^4^ (median (RIQ))	188 (140–256)	202 (143–270)	185 (134–249)	0.522

^1^ CNA: *Candida* non-*auris*. ^2^ SMD: Standardized mean difference. ^3^ SD: standard deviation, ^4^ PaO2/FiO2: ratio of arterial partial pressure of oxygen over the fraction of inspired oxygen.

**Table 3 jof-09-00715-t003:** The treatment and unadjusted outcomes in seven centres in Colombia, 2016–2021.

Outcome	All n= 512	*C. auris* n = 134	CNA ^1^ n = 378	*p*
Received Antifungal (%)	441 (86.1)	119 (88.8)	322 (85.2)	0.370
Previous (%) ^2^	54 (10.5)	22 (16.4)	32 (8.5)	0.016
Timely (%) ^3^	175 (34.2)	48 (35.8)	127 (33.6)	0.719
Late (%) ^4^	212 (41.4)	49 (36.6)	163 (43.1)	0.222
Total mortality (%)	288 (56.2)	66 (49.3)	222 (58.7)	0.072
Mortality at 30 days (%)	244 (47.6)	51 (38.1)	193 (51.1)	0.013
Immediate (%) ^5^	53 (18.4)	2 (3.0)	51 (23.0)	<0.001
Early (%) ^6^	122 (42.3)	19 (28.8)	103 (46.4)	0.016
Late (%) ^7^	122 (42.3)	32 (48.5)	90 (40.5)	0.315
Mortality after 30 days (%)	44 (15.2)	15 (22.7)	29 (13.1)	0.085
Hospital stay in days (median(IQR))	30 (18.8–49.3)	32.5 (18.8–53.3)	28 (18.8–49.0)	0.860

^1^ CNA: *Candida* non-*auris*. ^2^ They received antifungals before taking the blood culture. ^3^ Initiated antifungal between day 0 and day +2. ^4^ They started antifungal after day +3. ^5^ Death between day 0 and day +2. ^6^ Death between day 0 and day 7. ^7^ Death between day +8 to day +30.

## Data Availability

Data is available in an anonymous way, by request.

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
