# Peer review of "Mortality Caused by Candida auris Bloodstream Infections in Comparison with Other Candida Species, a Multicentre Retrospective Cohort"

_jof, 2023, doi:10.3390/jof9070715_

Round 1
Reviewer 1 Report
Overall a very interesting paper on the rise and pathogenicity of C. auris in patients with other comorbidities. Things that need to be addressed in the article starting with more scientific and finishing with wordsmithing and the like.
Line 38 why was the hospital stay not identified?
Line 41 Says no effect, how is that possible.
While reference 1 is fine it is old and before C. auris was identified as problematic. I think it would be more helpful to have a more recent reference
In a number of places C. auris and other species are not italicized. Example line 66, 219, 293
the writing in lines 60-73 is a little coppy and hard to follow.
The works cited needs to be checked for errors. Example reference 122 line 398
Given that Covid19 happened at the end of this study a comment addressing this would be informative to the reader. Also I would like to see a comparison of the effects of comorbidities, such as Bacteria vs Fungi vs Viruses for which combination had the most detrimental effects.
313 reference missing date
315 should state bloodstream
Line 329 What is meant by Drug-Resistant Candida? Is that a reference?
One thing I found lacking was a line on patient consent and/or an IRB
Minor issues with editing and with proofreading. A good readthrough and editing would help make the paper. I have noted some issue in the comments to the authors
Author Response
- Line 38 why was the hospital stay not identified?
Response: The wording was changed to reflect the fact that no differences were observed between the two groups, even after adjustment with inverse weighting.
- Line 41 Says no effect, how is that possible.
Response: As mentioned, the wording was changed.
- While reference 1 is fine it is old and before auris was identified as problematic. I think it would be more helpful to have a more recent reference.
Response: A new reference was put instead of the old one.
- In a number of places auris and other species are not italicized. Example line 66, 219, 293.
Response: Thank you for your comment. We made the adjustment throughout the manuscript, including the references.
- The writing in lines 60-73 is a little copy and hard to follow.
Response: The writing was simplfied to facilitate the reading.
- The works cited needs to be checked for errors. Example reference 122 line 398
Response: All the citations were checked and errors found in several references were corrected.
- Given that Covid19 happened at the end of this study a comment addressing this would be informative to the reader. Also I would like to see a comparison of the effects of comorbidities, such as Bacteria vs Fungi vs Viruses for which combination had the most detrimental effects.
Response: The following was added: “Part of the cohort included patients with COVID-19. However, the effect of this variable was not visible, since this characteristic was included in the regression model used to contruct the propensity score. In the pseudopopulation, no difference was noted in those with or without COVID-19. Neither did we collect data on bacterial or viral coinfections.
- Line 313 reference missing date
Response: Thank you for your comment. We made the adjustment
- Line 315 should state bloodstream
Response: Thank you for your comment. We made the adjustment
- Line 329 What is meant by Drug-Resistant Candida? Is that a reference?
Response: It was a typo, and the sentencewas corrected. It reads:”… treatment and impactfully address the current emergency of drug-resistant Candida species
- One thing I found lacking was a line on patient consent and/or an IRB
Response: Thank you for your comment. However, in the “Institutional Review Board Statement” and “Informed Consent Statement” sections (Lines 413 to 418) we mention that the institutional boards of all the centers made an exception/waiver regarding informed consent, because the retrospective nature of the study.
- Minor issues with editing and with proofreading. A good readthrough and editing would help make the paper.
Response: A complete proofreading was done.
Reviewer 2 Report
General Impression
The authors describe the results of a study on the comparative mortality of Candida auris and Non-candida auris bloodstream infections conducted in Colombian Health Centers between 2016 and 2021. The study concludes that the mortality of patients with C. auris infections is somewhat lower than for individuals infected with non-auris strains. The rationale for the study is clearly explained, the methods are described in appropriate detail, the statistical analysis of the data is sound, and the conclusions are appropriate. In particular the statistical correction for confounding variables through weighing of exposure propensity scores is well-executed, allowing the authors to avoid some of the most pervasive confounders of studies of candidemia mortality like diabetes or prior antibiotic use.
Suggestions for improvement
- It is lamentable that antifungal resistance data could not be obtained (lines 120 and 326) as they could hold the key to the observed discrepancy with similar studies. Is it possible that the comparably high mortality in the non-auris group is due to the high prevalence of multidrug resistant C. albicans in the area. Is there any chance that proxy data could be included?
- This reviewer would be interested in the factors that were identified in the multivariate regression analysis (paragraph starting line 153). Would it be possible to discuss which of the 27 factors in the propensity score calculation contributed most of the risk?
- English language is fine, few typographical errors should be evident during copy editing (e.g., reference 15 is incomplete).
Author Response
- It is lamentable that antifungal resistance data could not be obtained (lines 120 and 326) as they could hold the key to the observed discrepancy with similar studies. Is it possible that the comparably high mortality in the non-auris group is due to the high prevalence of multidrug resistant C. albicans in the area. Is there any chance that proxy data could be included?
Response: Unfortunately no data was available to adjust or improve the possible interpretation. However previous data of susceptibility among C. albicans and C. parapsillopsis isolates suggest a low prevalence of this problem (Nucci M, et al. PLoS One. 2013;8(3):e59373. doi: 10.1371/journal.pone.0059373).
- This reviewer would be interested in the factors that were identified in the multivariate regression analysis (paragraph starting line 153). Would it be possible to discuss which of the 27 factors in the propensity score calculation contributed most of the risk?
Response: No, since the regression was done on the probability of a C. auris isolate, no data on the effect of the vaariables included in such model over the mortality is available.
- English language is fine, few typographical errors should be evident during copy editing (e.g., reference 15 is incomplete).
Response: The complete text was reviewed for typos and clarity.